# Communitas, Worship, and Music: Using Music to Revitalize the Post-Modern Church

**Joshua Taylor**

College of Music, University of North Texas, Denton, TX 76203, USA; joshua.taylor@unt.edu

**Abstract:** Music often facilitates the experience of communitas within disparate groups of people. As the American mainline Protestant church faces schism and struggles for relevance in a post-modern era defined by mistrust in the institutional church and social subjectivism, reexamining how singing together can break down barriers within ecclesial structures and create shared understanding is merited. As demonstrated through the music of pilgrimage, community musicking allows individuals to define the sacred together. Music then becomes an educational resource for the reformation of the church. The Iona and Taizé communities offer insights into this process. Their publishing efforts and worship styles, influenced and crafted by the populations who visit their locations, have provided resources for this dialogue in localized contexts. However, the experience of communitas is individualized—no one person, group, or organization can define this outcome. Consequently, no single musical or liturgical approach will be appropriate in all contexts; the church's music must adapt so that each selection is imbued with meaning for that community. Facilitating such a process in the local congregation may threaten the status quo while also becoming a tool for revitalization in the post-modern era.

**Keywords:** church music; pilgrimage; communitas

## 1. Introduction

As the American mainline Protestant Church faces declining membership, lower worship attendance, and increased schism in the 21st century, the recovery of pilgrimage, and its music, may lead to a new praxis within post-modern congregations. Post-modern congregations may discover new understandings and deeper connections within their worship and common life through the adoption of what the author calls a "pilgrimage theology" and the music of pilgrimage. The liminal experience of pilgrims journeying together may create communitas among participants resulting in a new shared reality (Turner 1969).

Change for a church adopting a pilgrimage theology is more a perpetual time of transition. Informed by the past and looking toward the future but rooted in the present, communities develop communitas through their shared commitment to and respect for connection with one another and authenticity within their shared experience (Thomas et al. 2018). This perpetual transition represents the community's own metaphorical pilgrimage—a journey that faith communities often fail to acknowledge. The community's decision to turn from a "that's the way we always do it" mentality toward an evaluation of local traditions may incorporate more pilgrimage practice into their existence. Just as the experience of pilgrimage was discouraged by the historical church, present congregations adopting this "pilgrimage theology" may experience challenge to the church's existing structures and status quo (Kubicki 1999). For the church embracing a pilgrimage mindset, this challenge to their authority and structure is not unexpected.

Moving away from certitudes toward a more flexible approach and understanding of community is essential for the pilgrimage church. Movement away from a static theology and way of being is embedded within the pilgrimage discipline. The Iona and Taizé

communities have provided important examples for how this work may be accomplished in specific contexts (for example, Brother Roger's commitment to living in the provisional at Taizé) which are discussed later in this article (Hawn 2013). The music, prayers, worship experiences, times, locations, rehearsals, and outcomes must be amendable in the name of the individual developing a closer relationship with God. Movement—whether physical or not—is at the heart of this approach. In the post-modern era, the physical nature of pilgrimage is no longer an essential element for this approach (Taylor 2022). Pilgrimage requires the church community to be light on its feet and ready for change (Taylor 2022).

The (current) church leadership's willingness and ability to reform ecclesial practices including music offers one way to connect with individuals (both inside their congregation and among the growing number of individuals identifying as "spiritual but not religious"), potentially reversing the declining trends of the post-modern congregation (Lipka and Gecewicz 2017). The reformation of musical practices in the local congregation must focus on the shared meaning developed between a disparate group of individuals (Rieger 2015). Music has played a significant role in shaping the theology and identity of communities since the beginning of Christian worship (Kimbrough 2006). Similarly, music has been one way that pilgrims assign meaning during and after their journey (Bohlman 1996b). Moving beyond music for entertainment purposes, a uniquely American ideal (Myrick 2018), to a catalyst for meaningful shared experience among communities provides one path for this revitalization (Kubicki 1999).

For faith communities, embracing this pilgrimage approach requires interacting with, valuing, and incorporating the stories, traditions, music, and perspectives of all individuals. This creates what anthropologist Victor Turner calls a "flowing process" between constituents rather than a top-down approach (Turner 1977). By constructing shared meaning from individual musical preferences those of the existing community and the individual pilgrims a newly formed, or reformed, community has singularity of vision and common purpose by creating new practices outside of normal ecclesial structures (Coleman 2004). These groups' shared definitions of meaning and communitas may lead to revitalized worship and music practices that reach beyond the denominational and stylistic differences which have defined many American churches in the post-modern era.

## 2. Revitalization and a New Outlook

The church's ability to adapt, adjust, and incorporate pilgrims' individual religious identities in their structures and practice is significant for its future life. As the American mainline Protestant church continues to change in the 21st century, different approaches, understandings, and practices will be necessary for its continued relevance and survival. Less Euro-centric, Western traditional elements and more communal and/or folk practices such as pilgrimage may become necessary guideposts for the established faith community's future (Thomas et al. 2018).

The established congregation must work to find a balance between individual experiences of transformation and empowering a sustainable, workable community of individuals. Post-modernity has shown that individuals are choosing to belong to fewer organized communities with greater emphasis on individual expression (Rieger 2015). Incorporating these individual identities, including music, prayer, and rituals, alongside established ecclesial structures and orthodoxy, will take time, study, and intentional actions. This will likely result in churches reconsidering the location, times, substance, and results of their programming, including the music ministry, worship experiences, meetings, and other aspects of their community.

The adoption of a pilgrimage theology requires churches to consider the pilgrimage community first. Whereas membership in the church community was once defined by individual members fitting into these pre-defined structures, post-modernity and the adoption of a pilgrimage theology become a collective process. The pilgrim plays a significant role in the definition of what is sacred for the community challenging the homeostasis of the established organization. The faith community's commitment to individual development

and a shared collective meaning requires them to adapt to different individuals' perspectives. This does not mean that conflict is removed from the church experience; rather, it is suspended for the purposes of communitas (Thomas et al. 2018).

Music may provide the best opportunity for dialogue in this collective process. For the pilgrimage church, former understandings of their life together are not erased; rather, the community is one of compromise and adaptation (Sallnow 1981). Like the ways in which the Iona Community has used new hymn texts to speak of sensitive and often divisive social issues, the music of the church may open a dialogue about challenges between the church's orthodoxy and the individual pilgrim's context of meaning and social outlook. The post-modern church's ability to harness the individuality of pilgrims for their own success is dependent upon acknowledging the pilgrims' identity. Like pilgrimage, discipleship may happen in the context of a community and be shaped by that community's structures, compromises, and identity, but it remains an individual undertaking. In adopting pilgrimage attributes, it is necessary to never lose connection with the individual even if the individual's perspective may not always be positive or conflicts with the existing church.

Faith communities' recognition that maintenance of the status quo is no longer an option is critical when considering the pilgrimage discipline as insight for a different future. Just as the pilgrim is "at home in their sojourning", the pilgrimage church must recognize that incorporation of individual constructs of meaning, shaped within community, for the purposes of transformation is a critique of current practice. Organizations reflecting on the experience of the pilgrim may be encouraged to consider what practices and adaptations to their life together might produce their own liminal moments. Allowing space for a sense of liminality to pervade their spiritual and organizational understandings also invites a wider conversation about what is happening within their community (Power 2006). For American mainline Protestant Christian churches, this liminality also produces a deeper understanding and recognition that God is at work not just in their community but in communities around the world. The dissemination of Taizé music around the world demonstrates how global practices may inform local practice when congregations open themselves to the experiences of pilgrimage (Kubicki 2013). The re-discovery of the pilgrimage discipline for a post-modern church requires the church to embrace an active theology embodied in discernment of individual meaning, common practices including prayer and music, and a willingness to move beyond the status quo (Taylor 2022).

## 3. Liminal Space, Communitas, and Music's Role in the Post-Modern Context

An understanding of communitas and pilgrimage theology invites American mainline Protestant congregations to consider the liminal space necessary in worship and praxis to subvert the schism and divisiveness present in the 21st century. In their ground-breaking studies on the ritual of pilgrimage, 20th-century anthropologists Victor and Edith Turner defined the term communitas as being "the intense sense of togetherness, often experienced in ritual experiences, where a group stands outside of normal societal structures" (Turner and Turner 1978). Building on Arnold Van Gennep's theory of liminality, Turner and Turner suggested that the departure from the safety of home, the trials of travel, and the enlightenment of the journey, create a liminal space in which individuals form a new community, and thus new societal structures, because of their shared pilgrimage (van Gennep et al. 1960).

Music is one means for creating this liminal space. Musicologist Philip V. Bohlman argued that there is no situation in which music does not either amplify or create the conditions necessary for this new bounded world (Bohlman 1996b). Musicking (Christopher Small's term that encompasses all acts of musicmaking) provides one significant way that pilgrims form community, respond to the journey, and work for the transformation that motivated their journey (Small 1998). The songs of their journey become an expansion of their individual pilgrimage stories that impact both the temporary community of their pilgrimage and their communities back home. Whereas ethnomusicologists have

historically focused on the role of music as mediator between sacred space and physical space (Ingalls 2011), this limited focus fails to recognize the role of music and ritual in the overall pilgrimage experience (Bohlman 2001). Anthropologist Simon Coleman and sociologist Martyn Percy expanded the understanding of pilgrimage beyond its original physical nature to include the possibility of a metaphorical journey. As scholars Evgenia Mesaritou, Coleman, and John Eade noted, "The field of pilgrimage studies has greatly expanded in recent years... [while it is still concerned with the] labor, politics, power relations involved in the construction of sacred centers, but also the ways in which the field of study must be extended to other places where pilgrims learn to practice their religion, and live their everyday lives" (Mesaritou et al. 2020). This expanded definition provides more possibilities for understanding pilgrimage and music in the post-modern context.

Music's role on pilgrimage may be understood as a practice that informs the pilgrims' outlook throughout the journey and at the conclusion of their pilgrimage. Monique Ingalls suggested that "the 'sacred center' of post-modern pilgrimage comprises not specific places or physical objects, but rather portable practices and discourses intended to be 'transferable' back into more localized ecclesial contexts" (Ingalls 2011). The portable practices of post-modern pilgrimage, combined with the disconnection of pilgrimage from its spiritual moorings, has opened the discipline to individuals identifying as "spiritual but not religious" which could suggest the benefit of its reclamation by the 21st-century church (Lipka and Gecewicz 2017; Taylor 2022).

As pilgrims are formed by the journey, their worldview, including liturgical and ecclesial expectations, may change. Whereas individuals embark on pilgrimage for different reasons, their rationale, practice, and definition of the discipline shape their experience and that of the wider group of which they are a part (Collins-Kreiner 2010). This experience reforms their own story as well as the stories of their home communities when they return (or the community they've traveled to; see the Taizé example discussed later in this paper). For a metaphorical pilgrimage, the return home may be as simple as returning to the established worship order after a provisional time exploring other options.

Pilgrimage groups' intense feelings of togetherness and navigation of societal structures, the phenomenon defined as communitas, provides the impetus for transformation among the individuals of the group. The motivations, practices, and outcomes of these shared journeys have produced historical consequences for the individual, the pilgrimage group of which they are a part, and the community to which they return at the end of the journey. These consequences and the new realities created through historical pilgrimage provide a roadmap for a renewed understanding of the discipline in the 21st century.

A brief definition and history of pilgrimage may be helpful here.

## 4. A Broader Definition of Pilgrimage and Pilgrimage Studies for Consideration

A broader definition focused more on the practices of pilgrimage, whether physical or metaphorical, is necessary as it is reexamined in the 21st century. The word pilgrimage is derived from the Latin phrase per agrum, meaning "through the field", and the noun, peregrinis, meaning foreigner, wanderer, exile, traveler, newcomer, or stranger. However, these Latin roots fail to capture the practices, motivations, characteristics, and desired outcomes of the discipline.

In this expanded understanding, it is the practices, encounters, and interactions of these journeys that become the important features for study. Historically, pilgrimage studies have been focused on the geographical and religious features of the pilgrimage destination. Now understood in a more metaphorical sense, it is pilgrimage's ability to create new perceptions for the pilgrim and potentially create new expectations for the individuals upon their return home or within the established community that is of primary importance (Frank 2008). Anthropologists and sociologists have come to look at how the intentional actions of pilgrims direct their attention toward God with the hope of changing their current practices in the world and ultimately the communities of which they are a part

(Curran 2010). The development of the Taizé chants discussed in the next section provide an important example of this.

These intentional actions follow a predictable pattern. Dutch anthropologist Arnold van Gennep has described the pilgrimage structure as a rite of passage consisting of three stages: separation, ordeal, and reintegration (van Gennep et al. 1960). Turner and Turner, drawing on Van Gennep's definition of the pilgrimage cycle, understood the ordeal of the journey as being the locus for the development of communitas. In departing, the individual pilgrim is separated from their society as well as the structures of their religious institutions, creating tension between their motivations for pilgrimage, newfound freedom on the journey, and the rigor and orthodoxy of the local and ecclesial communities they have left (Socolov 2011). As they experience this tension, they are also shaped by the stories, identities, and motivations of the other pilgrims they encounter. It is this interaction that creates the potential for the development of a new reality. This phenomenon has been documented in the writings and practices of pilgrims throughout history.

Historical records of pilgrimage exist from the second century onward and offer insights into the discipline for its rediscovery. Pilgrimage, in its earliest forms, was outside of the church's normal orthodoxy, and it often remains beyond the norms of ecclesial structure frequently challenging the church's authority and the wider political and social landscapes. The earliest recorded Christian pilgrimage was in 170 A.D. by Milito of Sardis to Jerusalem with the peak of Christian pilgrimage occurring during the 12th and 13th centuries when it was considered a "major investment in eternal life" (Rieger 2015). The fourth-century Spanish nun Egeria, and, later, St. Francis of Assisi's writings detail the practices of pilgrims. Their writings offer glimpses into the specifics of the pilgrimage journey the other pilgrims they encounter, their liturgies and song, their interactions with the local communities of their destinations, and the church's (i.e., the established community's) response upon their return (Frank 2008). Historical pilgrimage led to the construction of roads and hospitals and shaped political realities (Pazos 2012; Tolan 2009), and 21st-century pilgrimage may have similar effects. In most cases, however, pilgrimage was discouraged by the church because of its ability to disrupt the established ecclesial order. These opinions were codified at the Council of Chalcedon which severely limited monastic travels (Frank 2008). Francis' writings, however, and the continued practice of pilgrimage suggests that, if pilgrimage had a downside, it was not apparent from the pilgrims' personal accounts.

In the 21st century, the ease of contemporary travel at least for many middle-class US citizens has made the sites of historical pilgrimage (and perhaps the discipline of pilgrimage itself) available to a greater number of people. Due to this, there is far greater diversity among pilgrimage participants than at any point in the history of the discipline (Cousineau 1998). Pilgrims now represent those with varying religious affiliations—or none at all. This diversity, along with the rising number of individuals embarking on pilgrimage, has led some scholars to conclude that it is the desire to rediscover the "down-to-earth realities of human beings" that makes pilgrimage a valuable resource in understanding community in the post-modern era (Pazos 2012).

As more pilgrims have traveled, more stories, practices, and experiences of their journeys have impacted their home communities. This has led to a greater focus on the ways in which pilgrimage might engage entire religious communities, reaching beyond the early focus on pilgrimage sites alone and the later attention to the motivations of individual pilgrims (Ingalls 2011). Whereas rituals along the pilgrimage journey might, at times, mimic or preserve the values of a pilgrim's home community, it is the liminal moments shared together on the journey that are full of creative and inventive possibilities (Kubicki 1999). These possibilities may potentially reshape entire religious communities. Communitas is realized through shared participation in common rituals during the journey. Diverse pilgrims may be united through common practices when connections of traditional relationships, faith backgrounds, and geographical locations can no longer be assumed (Rieger 2015). The complex narrative space created at the intersection of the individual pilgrim's motivation

and understanding of pilgrimage, the common rituals, and the community formed through music is the essence of what the discipline might contribute to the post-modern church (Bohlman 2001).

Musicking provides one significant way that pilgrims form community, respond to the journey, and work for the transformation that motivated their journey. Pilgrims' songs on their journey become an expansion of their individual pilgrimage stories that impact both the temporary community of their pilgrimage and their communities back home. Whereas ethnomusicologists have focused on the role of music as mediator between sacred space and physical space (Ingalls 2011), this limited focus fails to recognize the role of music and ritual in the overall pilgrimage experience (Bohlman 2001). Pilgrimage song reflects an important ritual and storytelling component that deserves further consideration as the present-day benefits of pilgrimage are considered. Again, the Iona and Taizé communities provide an example of this, which is discussed in a later section.

## 5. Music's Role in Mediating Pilgrimage Experience

Music has a multi-faceted role throughout the pilgrimage experience. Whether a physical or metaphorical journey, music serves as inspiration and a transformational mediator, storyteller, and community-builder along the way. Additionally, it is a reminder of the past, the connection to the pilgrim's home community and to established practice. Upon return or the completion of the pilgrimage experience, music becomes a way of sharing the new story and new expectations because of the journey.

Music on pilgrimage is both an individual and communal experience. The music of pilgrimage only carries meaning when it is performed by the community (Taylor 2022). This contextual meaning differentiates the music of pilgrimage from music in traditional liturgical practice and moves music's purpose and existence beyond worship entertainment a characteristic prominent in American Protestant worship, both mainline and evangelical (Mall 2018). It is also this characteristic that suggests music's potential for revitalizing the post-modern church. Musicking is one way pilgrims negotiate their individual quests for self-transformation and their relationship to and with one another (Wood 2014). Pilgrimage, like art and poetry, is concerned with meaning at each stage of the journey (Cousineau 1998). The music of the pilgrimage is one medium through which this meaning is created or transformed. Philip Bohlman suggested that this pilgrimage music, whether created on the journey, learned previously, or shared between pilgrims, defines the experience for the individual, is embedded in the practice of the community, and will help to re-tell the story upon the pilgrim's return (Bohlman 1996b). This music is not static but rather constantly changing as it is interpreted, shared, and recreated among pilgrims, the wider pilgrimage community, and others who reenact it upon their return (Bohlman 1996b). This ever-changing dynamic makes the music associated with pilgrimage an embodied act of faith.

Because the music of pilgrimage requires performance by the pilgrims, the traveling group must be able to actively participate. This is not only a pragmatic concern but also an act of hospitality as pilgrims come into community with one another. The inclusive nature of pilgrimage music requires structures and forms that are conducive to a wide diversity of performers. The repertories are frequently multilingual and adaptable to the specific contexts and traditions of the pilgrimage group (Bohlman 1996b). Cyclical songs, often short and available in multiple languages, make the music of pilgrimage immediately accessible to a diverse group of people gathering for the first time, though they are far from the only form included in this vast repertory. Additionally, certain pilgrimage routes have yielded specific songs for that experience. For example, the Lourdes hymn is sung my many different groups traveling to the same location. The specific characteristics of these songs may be explored further in Philip Bohlman's musicological studies of pilgrimage and Judith Kubicki's writings about the music of the Taizé Community, among others. See the next section for ways in which this is accomplished in the Iona and Taizé communities. Similar

musical practices may also be found in non-Christian pilgrimage experiences (Sallnow 1981; Hornabrook 2018).

Individual pilgrims use music as a vehicle by which their diverse cultural backgrounds and contexts meld into a shared experience. Like the pilgrimage discipline itself, this music and their musicking often falls outside the bounds of traditional religious repertories and canonical practices (Bohlman 1996b). This might be seen in the groups who travel to Iona or Taizé. The worship forms and repertoire of these locales rarely match the home community of the individual. The pilgrims' musical genres and sources are diverse and consist of devotional music brought from home, the music of others, and the experience of a wider soundscape in the destination of their journey (Wood 2014). The music of their experience is not overly concerned with the musical orthodoxy of the church but rather with the connection it creates between the diverse pilgrims.

The new communities formed through pilgrimage music combine elements of what has come before, the story of the journeys (literal or metaphorical), and the new song that is produced through the interactions of diverse pilgrims along the way. Whereas it may not be music alone that creates a new story for pilgrims, music can transmit meaning and understanding if approached from a new hermeneutical paradigm a new pilgrimage theology (Kubicki 1999). Although not specifically related to pilgrimage, Monique Ingall's research on the music of non-traditional worship gatherings demonstrates how the ideas and music outside of traditional worship is taken home and influences and shapes practices and beliefs at the local level (Ingalls 2011). Ingalls suggests that participants in these gatherings take the songs, ideas, and soundscapes of these gatherings home with them and attempt to replicate them in their own contexts (not always with successful results). This is essential to considering the ways in which pilgrimages and their music might be used in the reformation and revitalization of churches in the post-modern era.

The performative nature of pilgrimage songs and their connection to the social and political contexts of the present pilgrimage group make them relevant for post-modern discussion. The characteristics of any individual pilgrim song may no longer be of primary importance when considering the impact of this music its connection to the past, narration of the present journey, and potentially transformational value following the journey. This broader understanding suggests that any song might function as a song of pilgrimage if it narrates a shared history with previous pilgrims, is relevant to the current group, and allows for the experience of pilgrimage to be shared, at its conclusion, with others through its subsequent performance (Bohlman 2001). The use of South African Freedom Songs or songs from the American Civil Rights movement in mainline Protestant worship (and their inclusion in the printed hymnals of various denominations) demonstrates this possibility. Like their reasons for departing on pilgrimage, individuals bring their own musical identities to the pilgrimage experience shaped by their own spiritual past that, in turn, shapes their current experience and the retelling of that experience (Thomas et al. 2018). The music of pilgrimage becomes then an important narrator for the journey, connecting the history of "home" and the experiences of past pilgrims with the current pilgrimage and, hopefully, to a new transformed future.

Music becomes the tool by which pilgrim communities define themselves and their image of the "ideal community". Because of the inclusive structures of the music, all members of the group share in crafting that image. Relationships and meaning are defined through the performance of the songs on the journey (Stokes 1999). Since the music must be performed to have meaning, the shared performance offers insight into the intersections and compromising between contexts, meanings, and goals of pilgrims. Additionally, the music performed shows what communal values are shared, indexed, encouraged, and will be used to transmit the meaning of the journey upon the pilgrims' return to their respective homes (Ingalls 2011).

The desired outcome of pilgrimage is that the image of an "ideal community" becomes a guidepost by which the pilgrims approach all future encounters. The transformed individuals use pilgrimage songs not only as narration on the journey but also as com-

munication tools for sharing their new stories and reality as a result of their pilgrimages. As the individual members of these pilgrimage communities interact, define their values, and sing together, the hoped-for transformation of their journey becomes a communal event. Additionally, their shared music becomes a corporate expression of that transformation (Cousins 2017). Demonstrated through shared musical expression, this implies that pilgrims need connection with God and the community to experience transformation. If pilgrimages are to have a lasting transformational impact on the individuals, the communities of which they are a part must also transform. Music provides one means by which the spiritual ideas of pilgrimage journeys might intersect with daily life and worship upon return (Bohlman 1996b).

The pilgrims' songs have the potential to become a revitalizing tool for post-modern churches standing in the face of the status quo. Their wider repertoire, employed to include the songs of everyone, creates a new story that changes the pilgrims' outlook for society and their home communities (Hawn 2013). An individual's initial reasons for undertaking pilgrimage often illustrate the ways in which they are seeking something beyond the comfortable realities of their current world. The music of pilgrimage often demonstrates how the pilgrim's transformation may be dependent upon the broadening of their worldview or, possibly, the obfuscation of their current ideas (Bohlman 1996b). The eclectic encounters with "the other" and the formation of a new song through their shared experiences contributes to the broadening of view for the individual pilgrim, which may raise awareness of systemic issues and concerns about the practices of their home communities (Ingalls 2011).

## 6. The Iona and Taizé Communities: Examples for Post-Modern Pilgrimage

The Iona and Taizé communities provide important examples for how the interactions of pilgrimage can change whole communities. Interestingly, both communities offer the opportunity to study how an existing community has been changed by the presence of pilgrims, and the opportunity to understand how pilgrimages to these locales have changed congregations upon pilgrims' returns from them. In both cases, music played a seminal role in the transformations.

Founded in 1949 by Brother Roger Louis Schutz-Marsauche (1915–2005), the ecumenical community located in Taizé, France (hereafter referred to as the Taizé Community) has a long history of welcoming pilgrims from around the world and employing music as a means for extending hospitality, building community, and shaping transformative moments for their visitors. Although not originally established to be a site of pilgrimage, the thousands of individuals from four continents who travel to Taizé each year connect the Community to the pilgrimage discipline (Kubicki 1999). In his authoritative work on the early history of Taizé, J.L. Gonzalez Balado described Brother Roger's twofold vision to be "a community that prays, rooting all its life in contemplation, and a place where the generations meet, where the young are made welcome" (Balado 1981). Through their commitment to providing hospitality, the Taizé Community opened monastic practices to a wider, more ecumenical church (Santos 2008).

The brothers of the Community came to understand that their forms of prayer, music, living, and worship space would need to adapt to accommodate the large number of pilgrims. The brothers gradually substituted what had been sacred and traditional for them for that which was accessible to the wider group (Hawn 2013). As early as the 1960s, the brothers expressed a desire to utilize music in the prayers that would facilitate the inclusion of the diverse pilgrims in their worship (Santos 2008). The chants that developed were primarily focused on enabling the active participation of all in worship, not on creating a new musical or liturgical form, but on bringing young people from different backgrounds into a common form of prayer.

The specific form of the music used by the Taizé Community offers an example of a new form created as the result of pilgrimage. Meant to foster community among pilgrims from all over the world, the Taizé chants are short, repeated melodies in multiple languages

(Kubicki 1999). The songs at Taizé are taught in daily "choir practices" and are tailored to the needs of the pilgrimage group for any given week. For example, songs selected are often in the language of the predominant group present that week. The composition of the chants began in 1974 when Brother Robert, credited with the creation of the unique Taizé form, was preparing for Taizé's first Council of Youth. He realized he needed accessible music that would encourage the participation and did not assume any prerequisite knowledge. After the successful use of Michael Praetorius' canon, *Jubilate Deo*, Robert believed that other similar music might be used in the prayers. He held that this music must consist of "original compositions of solid quality that can be used by the people of God… and in this sense be called popular" (in Kubicki 1999). More in-depth information on the musical development of Taizé can be found in the writings of Michael Hawn, Judith Kubicki, Jason Brian Santos, and in the history of Taizé written by José Luis González Balado, a former brother of the community (Hawn 2013; Kubicki 1999; Santos 2008; Balado 1981).

The multiple languages employed, or the use of Latin by the entire assembly, allows the Taizé chants to serve as community-builder in worship creating a liminal space shared by the diverse assembly. Jason Brian Santos noted about this shared liminal space, this encounter with "our neighbor forces us to put aside our own pride and ideologies and focus on what is central to all Christians" (Santos 2008). In setting aside their own ideologies, the pilgrims are opened to a new worldview because of the Taizé music. These songs, encountered on their individual pilgrimage journeys, invite the pilgrims into a shared community through their active participation in the simple forms of the Taizé chants. The Taizé chants provide a freedom of worship expression that is known for its ability to create relationship among a diverse group of people while also allowing individuals to move toward God and transformation at their own pace.

The Community's form of worship and song has impacted not only the lives of pilgrims who have visited Taizé but also the worship lives of Christian communities around the world. In addition to visitors to Taizé taking songbooks home, Taizé's impact in the United States has been particularly profound due to their highly successful agreement with GIA Music Publications to market and sell their worship materials. As Michael Hawn notes about the impact of Taizé, "to many worshipers in the United States, prayer in the style of the Taizé Community with fewer words and extended periods of silence may be at once disorienting and refreshing" (Hawn 2013). With the music of Taizé now included in many American mainline Protestant denominational hymnals, the recreated story of pilgrims has had an impact on the practice of local communities. Regardless of its success, the experience of Taizé worship has caused pilgrims' home congregations to consider how their worship offers space for individuals to pursue God through silence and simplified musical forms embedding aspects of the pilgrimage discipline into their local liturgical practices. The adaptability of the Taizé Community, especially as seen in their preferred musical form, suggests the innate ability of pilgrimage to reshape the story of both the individual pilgrim's life and that of an entire community.

The impact of the modern-day Iona Community offers different insights into the pilgrimage discipline. George MacLeod's (1895–1991) social experiment, begun in 1938 and later known as the Iona Community, stands as the latest chapter in the long history of an island that has been a place of pilgrimage for centuries. From its earliest days, Iona was a location of religious diversity and a mixture of cultures. Due to this, it has always been a place where people of different backgrounds and lifestyles lived together negotiating their individual sense of freedom with the need to belong to the island community (Power 2006). MacLeod's experiment was rooted in the idea that Christianity was a community-based faith and that the church was failing to provide such a community. MacLeod believed that the individual was called to discipleship within the context of a supporting and demanding community and the current state of the church required a new approach (Ferguson 1998). From the beginning, the Iona Community has been rooted in community-building and accountability to one another.

Committed to diversity and radical hospitality, the Community adopted an ethos that favored questions rather than answers—a position that dramatically shaped the abbey's music and worship life. The group continues to be influenced by MacLeod's vision for a shared common life and his desire that Iona be a place for questioning and challenging faith that shapes the future lives of pilgrims once they return home (Ferguson 1998). With an eye toward a radical future, the Iona Community welcomes guests and seeks to build community among the many diverse pilgrims with the hope of influencing and shaping practices within the wider church.

The intersection of worship, work, and learning have been at the heart of the Community's understanding of themselves since the beginning. The music and liturgy of the organization, different in structure from the strident order of the Church of Scotland, have long attracted pilgrims to their work. Original liturgies have been developed to recognize the different backgrounds of the worshippers—an extension of the Community's commitment to hospitality. Weekly services focus on justice, peace, healing and include regular celebration of the eucharist (Power 2006).

The Iona Community views the music and liturgy of their services as an opportunity to extend hospitality to the diverse individuals who travel to Iona. Each element of the service is crafted to include individuals who come from varying faith backgrounds or none at all. This includes music. Whereas the Iona Community has contributed a vast number of new hymn texts and tunes to the congregational song repertoire, it is a "musical hospitality" and a commitment to including songs of the "other", selections from the global church, and new songs that supplement the traditional forms of church music, that ultimately defines their musical identity (Bentley 2009). Jane Bentley, a former Iona musician and community music scholar, suggested that the Iona Community's use of paperless music, instruction at the beginning of each worship service, and cyclical song provides an important example of this practice (Bentley 2009). Similar to the Taizé Community, the Iona Community has discovered, over the decades of its communal life, that this commitment to the inclusion of all people in the music of their services and the use of diverse music to form community has necessitated the composition of new resources and adaptation of their worship practices.

The creation of new musical resources for the Iona Community has largely been the work of John L. Bell (b. 1949), Graham Maule (1958–2019), and the members of the Wild Goose Resource Group, a semi-autonomous project of the organization. Bell, a Church of Scotland minister, became the prominent figure promoting the group's extensive song collections, anthems, shorter songs, and liturgies around the world. Bell's writings and workshops, particularly in his books *The Singing Thing* and *The Singing Thing Too*, have dramatically influenced the worship life and musical approach of the Iona Community (Hawn 2013). However, he was quick to say that the music and this ethos were established through the communal efforts of the Wild Geese. Bell stressed the communal nature of their creation, saying, "The songs from Iona are not composed in solitude on the beach. [They are] the product of ongoing argument, experiment, study, discussion, questioning, and listening to the conversation of ordinary people" (Hawn 2003).

The ever-changing pilgrimage group and the Iona Community's commitment to peace and justice, diversity, and singing the song of the "other" means that the musical resources of the organization's worship continue to expand and change. As Jane Bentley asserted, "music is seen not just as a tool for expression, but also for growth" (Bentley 2009). It is with this in mind that the Iona Community approaches music as a way of shaping the story and beliefs of the gathered assembly. It remains a part of the abbey musician's job to "ensure that music in the services challenges and expands the horizons of participants" (from the author's job description while serving as the Iona musician).

Like the Taizé Community, the Iona Community's prolific publications shape the discourse of communities well beyond the island of Iona. More information about the Iona Community's ethos and musical development can be found in Ronald Ferguson's history of the Iona Community and the many publications of the Wild Goose Resource Group.

## 7. Pilgrimage's Impact and the Post-Modern Church

Communal understandings become the dominant feature in pilgrims' post-journeying narrative. Although it is true that individual pilgrims undertake the discipline for different reasons, pilgrimage does not happen in isolation. Eurocentric ideas of family, faith tradition, and culture can no longer be assumed but, poststructuralist theories suggest that meaning is not fixed but shaped in relationship (Rieger 2015). Drawing on the diversity of the pilgrims gathered, participants write and craft liturgy and song together on Iona. Similarly, at Taizé, the singing of the chants following evening prayer often lasts late into the evening with the gathered pilgrims calling out the numbers of songs to sing, and the group indexing what is meaningful by which songs are sung each night.

The pilgrim is a stranger among strangers negotiating a new story and practices over the course of their shared journey. Pilgrims construct the meaning of their journeys together and through experiences with one another (Bauman 1996). These common experiences may include the sharing of songs, texts, liturgical practices, and other customs from their local communities as well as new rituals created or discovered along the way. Whereas pilgrims are unified in their common direction and in the process of creating a new story, their journey is still rooted in the historicity, orientation, and identity of the community(s) from which they came (Brueggemann 1977). As preeminent pilgrimage scholar John Eade noted, "the invention of ritual and the creation of sacred space raise the issue of authority... the rituals and the creation of sacred spaces is dependent on the ritual experts. However, the participants (pilgrims) also have agency...we see the increasing agency of the ritual participants, who brought their own knowledges, questions, and consumer items to record their experiences" (Eade 2020). It is by giving up the safety of their own social and ecclesiastical communities that individuals develop bonds with others—often from different social, economic, and faith contexts—while on pilgrimage.

Choosing to enter this communal experience, Christian pilgrims recognize that Jesus Christ also came as a stranger and, in so doing, question political, liturgical, and theological practices along the way. Presbyterian pastor and author on pilgrimage practices, Paul Lang, suggested that a Christological understanding of pilgrimage demands that pilgrims view each interaction with a stranger, new song, or unfamiliar ritual as a potential encounter with the risen Lord (Lang 2019). This perspective has ramifications for faith communities as they become increasingly ecumenical and diverse in the 21st century.

For pilgrims, the experience of traveling together brings them into solidarity with one another, forcing everyone to give up some control. The same may be necessary for ecclesiastical structures, influenced by pilgrimage, in post-modern society. The pilgrimage experience often demands a renewed energy and commitment to an active theology and robust liturgical practices, including music, that challenge social issues and push the bounds of traditional church practice. Pilgrimage and its associated rituals, including music, engage an active theological exchange with pilgrims sharing and comparing experiences and theological perceptions (Rieger 2015). Congregations wishing to reconcile the experience of individual pilgrims will have to confront the potentially static nature of their buildings, established worship orders, music ministries focused on performance, and other deeply held traditions. The pilgrim will expect that their journeying will also produce a new, meaningful reality for the established community.

As explained previously, music presents one way to allow space in worship for potential liminal experiences needed to explore these social situations and constructs. Community-based music that requires no prerequisites may allow for the Holy Spirit to work through an experience in which the community is temporarily equal, having come together for a singular purpose—like the pilgrimage experience (Collins-Kreiner 2010). The development of communitas, the acknowledgement of individual human identity, and a move toward greater universality and unity within the structure of a faith community will require new approaches in liturgical and musical practice.

Developing mutual relationships among pilgrims, with their individual constructs of meaning and established patterns of community, will take effort from the established church

and the pilgrim alike. The very nature of the pilgrimage theology is one of negotiation and mutuality. Individuals become a part of a larger whole through participation in common rituals and music which enhance group cohesion (Kubicki 1999). Additionally, pilgrims enter the discipline following in the footsteps of those who have undertaken sacred journeys before them. These realities suggest that the formation of community on pilgrimage and in the local faith community is contextual and relational. Congregations choosing to engage in a wider understanding of the pilgrimage discipline and its implications for their own practice need not think it necessary to change their entire way of being but rather be open to the diverse experiences that individual pilgrims bring to the conversation. As Marion Bowman put it in her study of new pilgrimage routes in Scotland (post-modern pilgrimage may be the place), "whereby material culture, places, and praxis from the past are (re)presented as meaningful in the presence" (Bowman 2020). The practice of the community must be dynamic and open to redefinition as they seek to build community to further their ministry (Russell 1993).

The diverse experiences and music of pilgrimage link together the shared quest of the individual and the established community in a tangible way. The import of practices, songs, and understandings from the journey enable a wider worldview and reformed story for faith communities. Just as pilgrims learn songs, rituals, and different theological ideas from fellow travelers and others along their journeys, their home communities have something to glean from pilgrims' encounters. It worth restating that pilgrims experience freedom in their detachment from their communities with space to meet new people, hear new things, and question their assumptions. For the congregation adopting the pilgrimage approach and mindset, a similar disconnection from their status quo may aid the development of communitas and facilitate a different perspective with the hope of broadening the entire community's story (Collins-Kreiner 2010).

## 8. A New Story and Paradigm for the Post-Modern Church

As pilgrimages continue to expand and with greater globalization, different ways of understanding society and shaping it through music, worship, and church structures will be required for the faith communities to remain relevant. On pilgrimages, diverse individuals break down the boundaries of society and faith. The new story and new reality of these pilgrimages requires the faith community to reach beyond itself, opening itself to a wider worldview through which the pilgrims may pass (Bohlman 1996a). Whereas current research stresses the importance of the individual pilgrim's narrative, it is the wider impact and ramifications of these stories that are most significant for local faith communities in the American mainline Protestant context (Collins-Kreiner 2010). For those assemblies embracing pilgrimage theology (for example, the inclusion of individual stories, openness to reform in their current practices to include wider diversity of perspectives, and a commitment to dialogue about common definitions of what is meaningful and applicable to the present community), the hoped-for conclusion of pilgrimage is a new reality with practices and energy connected to and shaped by these individual stories. This may mean reaching across dividing lines of denomination or cultural identity to find commonality or including the perspectives of individuals identifying as "spiritual but not religious" into the common life of the congregation. For congregations engaging in this pilgrimage theology, seeking a way of being church that joins individuals together without negating their individual identities is the expectation. This way of doing life together requires a new reality for the established faith community.

For these communities, embracing pilgrimage ideals may require the congregation to become strangers within their own practices for the purpose of transformation. As with the cycle of pilgrimage for individuals, the faith community may have to embrace disorientation within its common life to allow the liminal space necessary to inform this new reality. This process may not be suitable for all congregations and will require a collaborative effort between both professional and volunteer participants. Additionally, it may not be necessary for assemblies to enact all aspects of the pilgrimage discipline but

rather be open to the way certain elements of a pilgrim's journey are impacting their life together. A focus on the individual narrative and those voices that were marginalized or silenced in the established order during times of social upheaval, technological advances, and greater ecumenicity will ask communities to acknowledge their own metaphorical pilgrimage journey (Ruf 2007). These moments broaden the community's horizons opening them to features important to the individual pilgrims.

If pilgrimage is to be re-discovered and valued in the post-modern church, the themes, practices, and music that pervade the stories of pilgrims upon their return should also shape the reality of the communities of which they are a part. Pilgrims' stories, rituals, songs, and other types of content suggest what holds meaning for the diverse pilgrimage population (Thomas et al. 2018). For faith communities, embracing this content in the contexts of church governance, liturgies, and musical practices will likely cause discomfort. However, it may be this reformation of church practices that becomes the catalyst for a renewed life together. The pilgrim's individual story, brought back from their journey, is the final part of the pilgrimage experience, and a community's failure to accept it or incorporate it into their own way of being potentially negates the possible benefits of the discipline and represents the maintenance of the status quo.

Embracing pilgrimage practices and ideals asks the faith community to look critically at their current practices and, possibly, move beyond their normal comfort zones. The potential benefits of pilgrimage for the post-modern church are only realized when the faith community acknowledges that choosing to walk with the pilgrim will create work and challenge for their community. Incorporating the practices of pilgrims into the church's life likely requires that the church look critically at its music, liturgy, governance, education, and way of relating to one another to include that which holds meaning for each individual. This incorporation must reach beyond an over-sentimentalized, romantic, or exotic view of pilgrimage to one of deep contemplation and commitment to understanding the discipline's benefits to the local assembly. Musically, it is likely that understanding pilgrimage will require the songs of the church to stretch beyond the dichotomies of traditional versus contemporary, western versus non-western musical idioms, and, potentially, beyond the realms of sacred and secular. Similar considerations are also necessary in relation to visual arts, architecture, governance, and language, among other areas. Giving up control in polity, liturgy, and music in the name of more widely embracing shared definitions allows space for the work of the Holy Spirit to create meaning beyond the church's efforts to define God only through their previous understandings (Lang 2019). Historically, the pilgrims' simplicity of dress, meals, accommodation, etc. while journeying suggests that living and operating in an alternative way creates meaning and value. However, the pilgrims' journey is also marked by uncertainty and a general lack of control. On pilgrimage, pilgrims give up their social, political, and ecclesial moorings in hope of transformation. This suggests jarring consequences for communities embracing a pilgrimage theology.

Just as the pilgrimage community finds liminal space in shared rituals, churches may also discover a new reality through the examination of their current practice and an openness or re-discovery of different rituals which unite individuals (Pazos 2012). Communities might rediscover their identity as much as change it. It is important to remember that the current pilgrim's experience is also connected to the experience of previous pilgrims rooting any transformation in a shared historical narrative.

Music has the potential to play a significant role in all this discernment. Attention to and a wider embrace of diverse musical practices encourages the formation of community when everyone's song is included. Acknowledging and naming this practice brings an awareness to a new way of being as a result of pilgrimage. By including their song, the pilgrim relives their community experience on pilgrimage and brings it into the collective conscience. Now a shared practice, the pilgrim's definition of what is sacred is allowed to permeate the collective conscience of the entire group.

The substance and practicalities of embracing these pilgrimage ideals, however, reaches beyond the fluidity of form or the simple inclusion of different music. As demon-

strated in historical writings, engagement in pilgrimage has had far-reaching political, social, and theological impacts (Pazos 2012). The individual pilgrims' new story may contain reformed ideas, subject matter, and expectations. What has been traditionally considered sacred may be re-discovered or discarded. The liminal space of pilgrimage has the potential to create new awareness, relationship with, and sacredness around people and issues that challenge traditional hierarchies. Victor W. Turner's research pointed to pilgrimage as a disruptor of processual units and societal structure. This insight suggests that the music of pilgrimage, and its other practices, may reach well beyond simple stylistic changes to fundamentally changing a community's identity (Kubicki 1999). On a larger scale, pilgrimage treaties, negotiations with political leaders, and the discipline's ability to transcend national and religious borders seems to indicate its ability to change whole societies (Stokes 1999). Regardless of scale, a deeper understanding of pilgrimage will require faith communities to examine the content of their worship and music in relationship to how it lines up with the pilgrim's (or the pilgrimage community's) desired societal outcome.

It is difficult to define the specific ways in which churches may need to stretch and dissolve these boundaries. Conceiving of alternative forms of structure will be uniquely contextual for each congregation but, most certainly, will vary from the current status quo with the hope of moving toward an experience of communitas between a socially, culturally, and economically diverse population. The connection to "heaven on earth" described by pilgrims demands an alternative physical world different from their previous situation (Rieger 2015). This reformed world moves beyond a comfortable, privileged, entertainment-driven worship and church reality. Examining the experiences of pilgrims, however, does provide some insight. Pilgrims frequently describe the development of communitas as the liminal moment when class distinctions fall away in favor of homogeneity among the pilgrimage community (Kubicki 1999). This connects the embrace of pilgrimage ideals with justice. For faith communities, embracing the pilgrimage discipline requires interacting with, valuing, and incorporating the stories, traditions, music, and perspectives of those beyond their walls, creating Turner's "flowing process" between the various groups (Turner 1972). For Christian pilgrims and faith communities, moving beyond the status quo has the potential to powerfully reshape the community and to acknowledge God's role in shaping the community beyond their own control.

Adopting a pilgrimage approach calls for the inclusion of many different perspectives in liturgical practice, polity, and song, among others. Already, greater opportunities for American travelers to be exposed to differing voices in the world church have shaped liturgical and musical practices in the church—as evidenced in the incorporation of Taizé and Iona music in mainline Protestant hymnals. The wider inclusion of global song and music from varying denominational and racial backgrounds suggests that traveling and individual stories or songs have become part of the Church's regular practice. However, other communities such as the Latinx community and the LGBTQIA+ community remain underrepresented or excluded. Rieger suggested that venturing further "off the path" may result in a "hybridity—the fused nature of identity that welds together the dominant and repressed" in one person or community. It bears repeating that this phenomenon requires the active incorporation of minority voices and perspectives as well as the relinquishment of control by the dominant community (van Opstal 2016).

As was often the case for the communities through which pilgrims traveled, change, because of pilgrims' experiences, is inevitable. With the ease of contemporary travel for most middle-class Americans and access to musical, liturgical, and spiritual resources via the internet, these narratives and practices are already informing the contexts of meaning for individual congregation members. For faith communities, these constructs may vary radically from the ideas and concepts of the (currently) dominant church (Rieger 2015). The ways in which churches can adopt and incorporate these ideas, stretching beyond their current status quo, may shape their relevancy in a post-modern society. Pilgrimage does not negate the previous world of the faith community but rather adapts it, creating an alternative and revitalized construct of that world.

## 9. Conclusions

In the post-modern era, pilgrimage affords the established faith community the opportunity to embrace individuals in their own quests for meaning while building authentic community through shared experiences, rituals, and music. This requires evaluation of their existing practices and a commitment to adopting a theology inspired by movement and sojourning. This theology is not new and is rooted in the historical and authentic past of the biblical narrative and previous pilgrims. The acknowledgment of this connection between pilgrimage's past, present, and future is essential for the complete realization of the discipline's ideals (Thomas et al. 2018). The quest for meaning and truth is one that reaches beyond time and tradition. Current pilgrimage joins this already rich story while accepting an unknown future.

The rediscovery of pilgrimages in the 21st century have much to offer church communities in American mainline church contexts. The Christian faith is rooted in the balance between individual discipleship and communal experience. Pilgrimage, defined by individual constructs of meaning and carried out in groups—either intentionally formed or formed along the journey—offers an important means for achieving this balance with potential ramifications for the post-modern church. The process of seeking the divine through pilgrimage and the practices of pilgrimage prayer, worship, and music among others potentially shape the meaning of any pilgrimage experience as much as the destination itself.

The practices of pilgrimage invite individuals to construct meaning on their own terms, and the communal negotiation of that which is best for the entire group creates a powerful connection or communitas and liminal moments among the pilgrimage travelers. This connection to one another allows the group to approach their destination from a place of vulnerability, an aspect of liminality, that hopefully facilitates transformation. The practices and community building aspects of the discipline potentially reshape the pilgrim's expectations both on the journey and in their return to the local faith community.

A pilgrim's or pilgrimage community's acceptance of something different than their status quo is necessary for growth. Pilgrimage, its practices, music, and rituals may expand the horizons of an individual or community's faith and provide greater understanding of their life-long spiritual journey, but it is likely to be gained through challenge to their previous existence. This alone may make the adoption and enactment of pilgrimage difficult for established faith communities. The imagination of the pilgrim is best captured when communities are able to move through this anxiety to an unknown sacred moment awaiting. Whereas 21st-century pilgrimage may differ greatly from Medieval journeying, pilgrimage today must reach beyond simple expectations that things will be easy, remain the same, or conform to the American Christian ethos. It is essential to the success of any pilgrimage undertaking to rediscover how the sacred reconfigures the concerns of pilgrims as opposed to the concerns of pilgrims reshaping the sacred. It is this readjustment that may lead to a re-examination of the established church's story upon the pilgrim's return.

The discovery of the sacred through pilgrimage will inevitably require a time of provisional and transitional approach to liturgical, theological, physical, and musical practices in the church. Pilgrimage, for the faith community, will be about learning as they go and rediscovering themselves along the way. Any re-examination of one's story, or a faith community's story, does not come without challenge. The moments of encounter with other people, music, worship, and other practices on pilgrimage provide a platform for this discernment connected to rich experiences. These encounters compel all involved, both those traveling and those who remain at home, to question their own assertions and ways of being. The attention to intention on pilgrimage produces opportunities to broaden one's horizons and open oneself to a new experience of the divine in ways antithetical to many other traditions in the current church.

This discipline may make a difference in the discussion of the post-modern church's relevancy. Pilgrimage may produce conversations around the church's purpose and a hoped-for transformation with a commitment to the valuing and incorporation of insights gained on the journey into the faith community's life together. How an individual or

community incorporates and reflects upon the practices of prayer, musicking, journaling, daily worship experienced on pilgrimage in their lives post-pilgrimage will ultimately determine the success or impact of the discipline in the long-term. Pilgrimage is about transformation of self and community through encounter. The hoped-for transformation of pilgrimage roots itself not in short-term gains but in a deeper understanding of everyday life, faith, and community. The cycle of pilgrimage includes a return, but it likely necessitates the giving up of life as it was known.

Ultimately, pilgrimage is rooted in tension, displacement, and the cycle of disorientation and re-orientation. These transformations often require ceding control and developing a more flexible approach, focused on the movement of the Holy Spirit rather than the maintenance of the status quo. The pilgrimage discipline overturns pre-conceived notions leading to the transformation of self for the individual and, potentially, for the reformation of entire communities and systems. Music is one connectional tool for this process. Singing the song of "the other" highlights the broader horizons to which pilgrimage beckons. This music, along with exposure to diverse worship practices, prayer, and encounters with people from differing backgrounds, traditions, and cultures, presents important transformative moments not only for the individual traveler but for the church and the world. This positions pilgrimage, with its music and other practices, as a significant resource for the evolving, reforming the life of the Christian Church today.

**Funding:** This research received no external funding.

**Institutional Review Board Statement:** Not applicable.

**Informed Consent Statement:** Not applicable.

**Data Availability Statement:** Not applicable.

**Conflicts of Interest:** The author declares no conflict of interest.

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
