# Peer review of "Communitas, Worship, and Music: Using Music to Revitalize the Post-Modern Church"

_religions, doi:10.3390/rel14091206_

Round 1

Reviewer 1 Report

I think the topic of this paper is very interesting, however, I don't quite understand a few things and there are several comments I would like to share with you:

1) If you refer to the church embarking on a 'pilgrimage journey' why do you bring in the Iona community as an example. There is no empirical data supporting your article at all.  I was hoping to read more about how Taize songs create the communitas experience. It would have helped to move beyond a pastoral theological approach. Those not familiar with Taize songs will have no idea of the sound or the lyrics of this unique type of congregational song. I think the incorporation of a short case study would benefit the argument of this article.

3) When you talk about current pilgrimage studies, it would be nice to reference some current studies as opposed to someone from 2000 (Collins-Kreiner 2010) on page 7. A valuable addition to this article could be Marion Bowman with her several articles on the burgeoning Protestant pilgrimage practices. Or the extensive work of pilgrimage studies doyen, John Eade. Or the author that you mention but only refer to his earlier work, Simon Coleman's latest book on pilgrimage (Powers of Pilgrimage 2021) would be essential to include and update the reference list. 

4) I suggest the use of the term 'musicking' by Christopher Small as opposed to music musicmaking.

5) The capital letters need a bit of harmonizing: e.g.: Cultural anthropologist, but earlier Anthropologist. Also, I suggest the lower case for century. 

6) Some of the references require your attention: when you say Turner & Turner suggests and you reference van Gennep on p. 1. It is also problematic when you refer to the works of Simon Coleman and Martyn Percy by referencing them through Ingalls 2011 article. It would be nice to read and reference the original authors. By the way, I recommend Ingalls' monograph  'Singing the congregation' published in 2018. Also, you reference Simon Coleman's 2004 work but it is missing from the bibliography. 

7) For me the anthropology of pilgrimage (studies by Turners, and Bowman, Coleman, Eade and others) and the concepts of pilgrimage theology is a bit blurred. There needs to be made a bit more reference to 'pilgrimage theology' and more discussion on how you relate the two to one another.

8) Some corrections need to be done in the bibliography. Including the works of Taylor. Kubicki 2021 is missing from the bibliography. Kubicki ND is also missing from the bibliography. You also need to look through your references of Turner. When you refer to their common work you need to reference them as Turner and Turner. And if it is only Victor, then it is OK to use Turner. There is a bit of confusion in the bibl. the publication date 72 then 78? Please, take a look at those. 

Author Response

Thank you for taking the time to read my submission. Here are responses to your questions:

1) I have added a section to the paper to provide specific examples from the Iona and Taizé Communities.

2) I have reviewed and added additional source materials including those referenced. 

3) I have updated to use the term musiking.

4) Thank you. All are corrected.

5) I have addressed these issues.

6) I have attempted to provide a clearer understanding of pilgrimage theology in the revisions.

7) The bibliography has been updated. 

Reviewer 2 Report

I have two major criticisms of this interesting paper. One is that the article would be more useful for readers if the author included an example or two of particular music. Though they say on line 201 that it is "beyond the scope of this article," it shouldn't be. A couple of paragraphs describing music of Taize or Iona communities, especially how they engender a pilgrimage/communitas experience, would sharpen and clarify the argument substantially.

Second, I found significant redundancy in sections 5, 6 and 7. Those sections should be condensed and possibly merged.

Regarding sources, the article really should reference Robert Wuthnow's important book, All in Sync: How Music and Art Are Revitalizing American Religion (2006). That book, now nearly 20 years old, addresses many of the same issues as this article. I would like to see the author engage it.

Also, in the very first paragraph the author cites a Kimbrough which is not included in the References.

Author Response

Thank you for taking the time to read my work.

I have added a section with case studies from Iona and Taizé to address your first point. Second, I have included the Wuthnow book in a revised version. 

Round 2

Reviewer 2 Report

Your comment mentioned you would be addressing Wuthnow's All in Sync in this revision, but I don't see it mentioned in the text or references.